# Advertising Innovative Sustainable Fashion: Informational, Transformational, or Sustainability Appeal?

**Valentina Carfora \* and Patrizia Catellani**

Department of Psychology, Catholic University, 20123 Milan, Italy
\* Correspondence: valentina.carfora@unicatt.it

**Abstract:** We aimed to understand how to promote innovative technology in the sustainable fashion market. The case study was the advertisement of a feminine bag with a chain coated using a new technology. We invited 550 women to read an Instagram post. In the control condition, the post only described the new technology. In the informational condition, the post emphasized the resistance and durability of the bag. In the transformational condition, the post emphasized the innovativeness and exclusivity of the bag. In the sustainability condition, the post emphasized the low environmental impact of its production. Results showed that the sustainability advertisement was the most persuasive in terms of consumers' involvement, systematic processing, and intention to buy the bag. In addition, reference to the functional benefits was an effective strategy to promote purchasing intention when consumers were interested in novelty and high quality, and when engaged in shopping for fun and enjoyment.

**Keywords:** innovativeness; fashion; sustainability appeal; transformational appeal; informational appeal



## 1. Introduction

Industrial manufacturing is changing the stability of the Earth system by crossing the environmental limits. Among the various sectors, the fashion industry is one of the most polluting. It is responsible for up to 10% of global greenhouse gas emissions linked to climate change, 20% of industrial water pollution, and 35% of oceanic microplastic pollution [1,2]. It is also the second largest water consumer and thus strongly contributes to its scarcity [1].

Given this scenario, the growing public attention and the international political commitment to environmental issues and sustainability have led to the development of a "sustainable fashion movement", which aims to remedy a variety of injustices in the fashion industry, such as environmental damage [3]. Over the past few years, the global sustainable fashion market has been increasing [4]. However, it is a still small market compared to the rest of the global fashion market [5]. Companies committed to the sustainable fashion movement attempt to reduce their environmental impact by adopting low-polluting and/or innovative raw materials and manufacturing processes, also increasing the physical durability of their products [6].

Starting from this scenario, in our research we focused on how consumers react to digital advertisements of accessories produced with innovative technologies that guarantee low environmental impact and prolonged duration over time. Innovative sustainable products can certainly be advertised online by underlying their pro-environmental features and, therefore, through a *sustainability appeal*. However, policymakers and brands may also employ the more conventional claims commonly employed in fashion advertising campaigns, namely informational and transformational appeals. While an *informational appeal* elaborates on product or service attributes or benefits, a *transformational appeal* elaborates on a non-product-related benefit or image [7–11]. To the best of our knowledge,

so far no scholars have compared these three categories of appeals to test, which is the most effective in increasing people's intention to buy an innovative and sustainable fashion product. In this study, we compared the persuasiveness of these three appeals in the online promoting of the purchase of an accessory produced with an innovative technology: a feminine bag prototype with a chain coated with physical vapor deposition (PVD) technology. Furthermore, we examined whether the effectiveness of these different appeals would vary according to the decision-making style of the consumer.

## 2. Theoretical Background

### 2.1. Informational, Transformational, and Sustainability Appeals in Online Fashion Advertising

The fashion industry plays a relevant role in the transformation of online advertising in new communication narratives mainly mediated by social media [12–14]. Among the various social media, Instagram is firmly positioned as a direct gateway between the fashion industry and consumers, e.g., [15], as it has become one of the most engaging social networks [16]. For this reason, understanding how to use Instagram to promote the purchase of an innovative and sustainable fashion product can be extremely useful. Although the means of communication have changed, the logic behind the creation of advertising content can still be linked to the traditional theorizing of the advertising appeal. Advertising appeal is the approach used to create branded content that attracts the attention of consumers and influences their feelings about a product, service, or cause [17]. An often-used distinction in advertising strategies is the one between informational (or rational) appeals and transformational (or emotional) appeals [18–22]. An informational appeal emphasizes one or more key features of a fashion product by highlighting the functional or utilitarian benefits deriving from owning or using that product [18]. The functional benefits usually correspond to the brand-related attributes [23]. These attributes can be classified into product-related and non-product-related attributes. Non-product-related attributes are the external aspects related to the purchase or consumption of the product (e.g., price, packaging, and product appearance). Product-related attributes are those aspects that are essential to make the product and are sought by consumers [24]. To emphasize functional benefits, the informational appeal focuses on facts, learning, and the logic of persuasion. Rational motives can be used as the basis for advertising appeals based on comfort and convenience, economy, health, touch, taste, smell, quality, dependability, and durability [18]. The informational appeal aims to impact potential customers' memory by creating a connection between advertising inputs and behavior [25]. It also stimulates evaluative thoughts, such as the source credibility and positive feelings [26].

A transformational appeal elaborates on an experiential benefit, that is, intangible characteristics such as how the brand makes you feel delighted and confident [27,28]. It may depict what kind of person uses a brand or what kind of experience results from buying a product [27,28]. Transformational advertising emphasizes the experience users will have when using the product and relates this experience with a unique set of psychological characteristics that would not typically be associated with it [29]. The aim is to evoke affective reactions that will motivate purchase [18]. Transformational advertisements can induce a wide range of emotional responses, from disgust to happiness [30]. As a matter of fact, transformational advertising is comparable to other psychological descriptors often applied to advertisement (e.g., mood, emotional, feeling, and image advertisements) in that it is essentially affect-based rather than cognitive-based [29]. An example of transformational advertising is the value-expressive appeal, also known as the image appeal [31]. This type of transformational appeal presents the social image that is attainable through the ownership or usage of a fashion product, underlying its positive consequences in terms of social status and reputation [32]. As with informational appeals, transformational appeals increase advertisement credibility and stimulate positive feelings.

Studies comparing the effects of informational and transformational appeals showed that emotional contents are better remembered and more effective than non-emotional

contents, e.g., [18,33], even if their effects seem to be less long-lasting [26]. In the context of fashion advertising, both appeals are widely used. In the scientific context, however, only one study has compared the effectiveness of these appeals in the fashion domain [34]. It focused on sportswear advertising and revealed that transformational advertising was more effective in increasing consumers' positive brand attitudes compared to informational advertising.

Besides the conventional distinction between informational and transformational appeals, with the advent of the sustainable fashion movement, we have started also to observe a massive use of the sustainability appeal. This type of appeal focuses on pro-environmental and ethical aspects of the fashion supply chain, such as reduced environmental impact of manufacturing, fair working conditions for employees, or monetary contributions of a brand to charitable causes [35]. The sustainability appeal has already been widely used in markets outside of fashion, and only in recent years it has become more common also in the communication strategies of the fashion industry. The relevance of sustainability appeals lies in the fact that they can increase not only sales, but also the consumers' attention and interest in the sustainability of fashion products, e.g., [36].

So far, only few experimental studies have compared the effectiveness of sustainability appeals with the one of informational or transformational appeals, and they have collected mixed results. Some studies showed that sustainability claims are effective in influencing consumers' intentions (e.g., [37]). For example, Yan et al. found that explicit messages on environmentally friendly products are better understood than other messages and enhance consumers' positive attitudes towards a brand, which, in turn, may shape their purchase intentions [38]. Similarly, a study on consumer perception of corporate social responsibility showed the potential value of promoting sustainable apparel with a sustainability message, by revealing that such a message enhanced the brand's corporate social responsibility image and, in turn, increased perceived brand innovativeness and consumer–brand identification [39]. In addition, Kim and Jin found that environmental advertisements combined with a rational or positive/negative emotional appeal increased purchasing intentions more than a neutral advertisement with no environmental appeal [40].

Other studies, however, showed a lower effect of sustainability appeals compared to conventional appeals. For example, Meyer concluded from his case studies that eco-fashion was perceived as less trendy than non-eco-fashion [41]. Likewise, Visser et al. showed that emphasizing an environmental benefit reduces buying intention through a lower perceived fashion image of the advertised product [42]. Given these limited and mixed results, we cannot exclude that sustainable fashion can be better promoted using conventional informative or transformational appeals rather than sustainability appeals. However, demonstrating the superiority of sustainable appeals would help convince more and more companies to use communicative contents that not only focus on retaining new customers but also on spreading a new sensitivity towards issues related to sustainable production. Therefore, in this paper we aimed to further investigate the effectiveness of informational, transformational, and sustainability appeals in increasing consumers' purchasing intentions.

*2.2. Consumer Decision-Making Style*

In addition to assessing which appeal can best convince people to buy innovative fashion products, it is interesting to assess whether each appeal can be differently effective depending on the type of consumer exposed to it. A long tradition of advertising research showed that consumers' psychological characteristics influence an advertisement's persuasiveness, e.g., [43,44]. They determine the extent to which a person evaluates the content as involving [45] and elaborates on it [46]. Consumers' decision-making styles are one of the psychosocial characteristics that can be related to their reactions to different types of advertisements.

Previous research has showed the existence of eight prototypes of decision-making style, e.g., [47–50]. Perfectionist consumers search for the best quality product and have

high standards and expectations. Brand-conscious consumers prefer the more expensive and well-known brands, being convinced that a higher price corresponds to a better quality. Novelty-fashion-seeking consumers continuously look for new things as they want to be always fashionable. Hedonic consumers shop just for enjoyment. Price-conscious consumers look for sales and prefer lower prices, being at the same time concerned about the value for money. Impulsive consumers do not plan their purchases, and they are not concerned about the money they spend or about the quality of their choices. Confused consumers have difficulty in choosing due to the presence of too many stores, brands, and products. Finally, habitual consumers have preferred brands and stores, and they remain faithful to them.

A consumer's choice can also be based on ethical decision-making, that is, a decision based on the consideration of whether a choice is right or wrong according to one's moral standards [51]. Therefore, a further prototype of decision-making style is the one made up by ethically minded consumers, who are concerned with environmental sustainability, social issues, health-related implications, and animal welfare [52]. Green consumers are a subcategory of this wider category and include those consumers who are especially interested in environmental protection [53].

Research has shown that decision-making style influences a variety of choices, including fashion purchasing. For example, hedonistic, impulsive, brand-conscious, and novelty-fashion-seeking consumers buy more apparel online compared to the other types of consumers. The same consumers (except for hedonistic consumers) also tend to spend more for purchasing fashion, unlike price-conscious consumers, who spend less compared to all the others. Moreover, price-, novelty-fashion-, and brand-conscious consumers have positive attitudes towards shopping using online social network sites, and they seek information about fashion products before buying them online [54,55].

As for hedonic consumers, their shopping decisions are based more on symbolic meaning than solid qualities, and a multitude of emotions play an active role in their purchasing experience. Hedonic consumers want to buy products that make them happy and prefer fashion products that they think will make them feel good when worn and that best suit their identity and the image they have of themselves. Prior studies have already demonstrated that transformational appeals influence the perception of hedonic and symbolic benefits related to purchasing the advertised product. They also appear to be very effective in communicating and enhancing subjective and emotion-based benefits [19].

Finally, as regards green consumers, several scholars have highlighted that their environmental attitude and concerns, as well as their perceived responsibility and pro-environmental values, are positively associated with sustainable clothing purchasing [56]. For example, Carfora et al. showed that attitudes towards buying sustainable clothing, and in turn the intention to buy it, were significantly predicted by a chain of values, beliefs, and norms related to sustainability [57].

Although the aforementioned studies have started to shed light on how consumers' decision-making styles influence their purchasing choices, to date no studies have considered how the same styles affect the effectiveness of advertising with different appeals. For this reason, in the present study we assessed the interaction between consumers' decision-making styles and the effects of different advertising appeals.

### 2.3. The Present Study

The main aim of our paper was to compare the effectiveness of informational, transformational, and sustainability appeals in increasing the intention to purchase sustainable fashion products, compared to a control. We exposed consumers to a social media advertisement promoting a bag prototype with a chain coated with PVD (i.e., physical vapor deposition) technology. While participants in the control condition only read appeals describing the PVD technology, participants in the informational, transformational, or sustainability conditions read appeals that also focused on the different qualities of the PVD

technology (e.g., increased resistance and durability of the product coated with it, novelty in the world of fashion, low environmental impact).

To evaluate the effectiveness of the appeals, we referred to past studies demonstrating that the more the message is involving and systematically processed, the more it influences consumers' intentions and behavior, e.g., [18,58]. Some message contents can be more effective than others in encouraging receivers to process the text as personally relevant to them, and therefore, they activate a higher message involvement [59,60]. In turn, consistent with the principles of the elaboration likelihood model [18], eliciting message involvement may lead individuals to engage in systematic processing, which implies a cognitive effort to analyze the content of the message and its relevance. Finally, this process leads individuals to behave in agreement with the recommendation presented in the message [18]. So far, no previous studies in fashion advertising have tested the differential effectiveness of informational, transformational, and sustainability appeals on message involvement, systematic processing, and, in turn, consumers' intention and willingness to pay. We therefore formulated the following general research questions:

Research Question 1 (RQ1)—Which advertisement (informational vs. transformational vs. sustainability vs. control) is perceived as more involving and processed more systematically?

Research Question 2 (RQ2)—Which advertisement (informational vs. transformational vs. sustainability vs. control) is more effective in influencing consumers' intention to buy and willingness to pay for a bag prototype with a chain coated with the PVD technology?

Our advertisements emphasized the durability and resistance (informational appeal), the innovativeness and social image (transformational appeal), or the sustainability (sustainability appeal) of the bag prototype. On this basis, we selected perfectionist, novelty-fashion-seeking, hedonic, and green consumers as the consumers' profiles that we expected to be most influenced by our appeals. In this regard, we formulated the following hypotheses:

**Hypothesis 1 (H1).** *Perfectionist consumers intend to buy the bag (H1a) and want to pay for it (H1b) more when they are exposed to the informational advertisement rather than the transformational or control advertisement. Their intention and willingness to pay are higher when they perceive the message as engaging and systematically process it (H1c). Perfectionist consumers also intend to buy the bag (H1d) and want to pay it (H1e) more than when they are exposed to the sustainability advertisement rather than the transformational or control advertisement. Their intention and willingness to pay are higher when they perceive the message as engaging and systematically process it (H1f).*

**Hypothesis 2 (H2).** *Novelty-fashion-seeking consumers intend to buy the bag (H2a) and want to pay for it (H2b) more when they are exposed to the transformational advertisement rather than the informational, sustainability, or control advertisements. Their intention and willingness to pay are higher when they perceive the message as engaging and systematically process it (H2c).*

**Hypothesis 3 (H3).** *Hedonic consumers intend to buy the bag (H3a) and want to pay for it (H3b) more when exposed to the transformational advertisement rather than the informational, sustainability, or control advertisements. Their intention and willingness to pay are higher when they perceive the message as engaging and systematically process it (H3c).*

**Hypothesis 4 (H4).** *Green consumers intend to buy the bag (H4a) and want to pay for it (H4b) more when exposed to sustainability advertisement rather than the other advertisements. Their intention and willingness to pay are higher when they perceive the message as engaging and systematically process it (H4c).*

## 3. Materials and Methods

### 3.1. Sample and Procedure

After the Ethics Commission for Research in Psychology of the Catholic University of the Sacred Heart approved this study, we invited 550 individuals to participate in the study through Prolific a platform for online subject recruitment designed for research. The study was advertised as a study on fashion product purchasing, lasting about 15 min. The inclusion criteria were being an Italian-speaking woman of legal age. We selected only women because the bag prototype was designed for women. After accessing the study on Prolific, participants provided informed consent through a questionnaire implemented using the Qualtrics platform.

At the beginning of the questionnaire, participants saw a fictitious Instagram post advertising a bag prototype with a PVD-coated chain. The post consisted of a video and a message. The video showed the bag with the PVD-coated chain and was accompanied by a persuasive message in the post caption. To investigate the effect of the message appeal on purchasing intention and willingness to pay, participants were randomly assigned to one of four different experimental conditions (using the automatic randomization function of Qualtrics): informational message, transformational message, sustainability message, and control message.

All participants received the following request: "To get started, we will show you an Instagram post promoting a bag with a chain handle. Please, take a good look at everything you see in the image and read the post caption carefully". In the control message, the post caption only described the PVD technology (Figure 1a). In the informational message, the post caption also emphasized the resistance and durability of a bag with a PVD-coated chain (Figure 1b). In the transformational message, the post caption underlined the innovativeness and exclusivity of a bag with a PVD-coated chain (Figure 2a). Finally, in the sustainability condition, the post caption highlighted the bag's low environmental impact (Figure 2b).

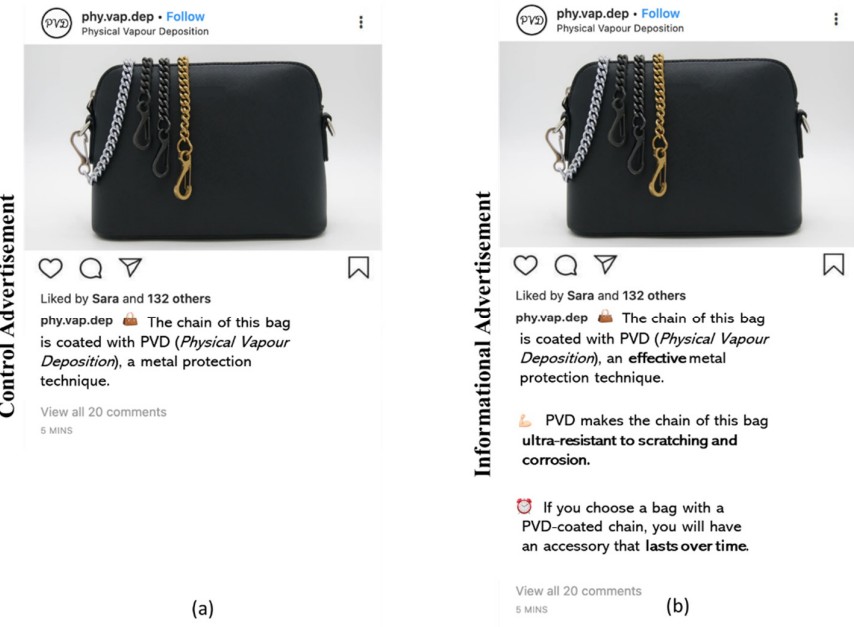

**Figure 1.** Instagram posts in the control (**a**) and informational (**b**) conditions.

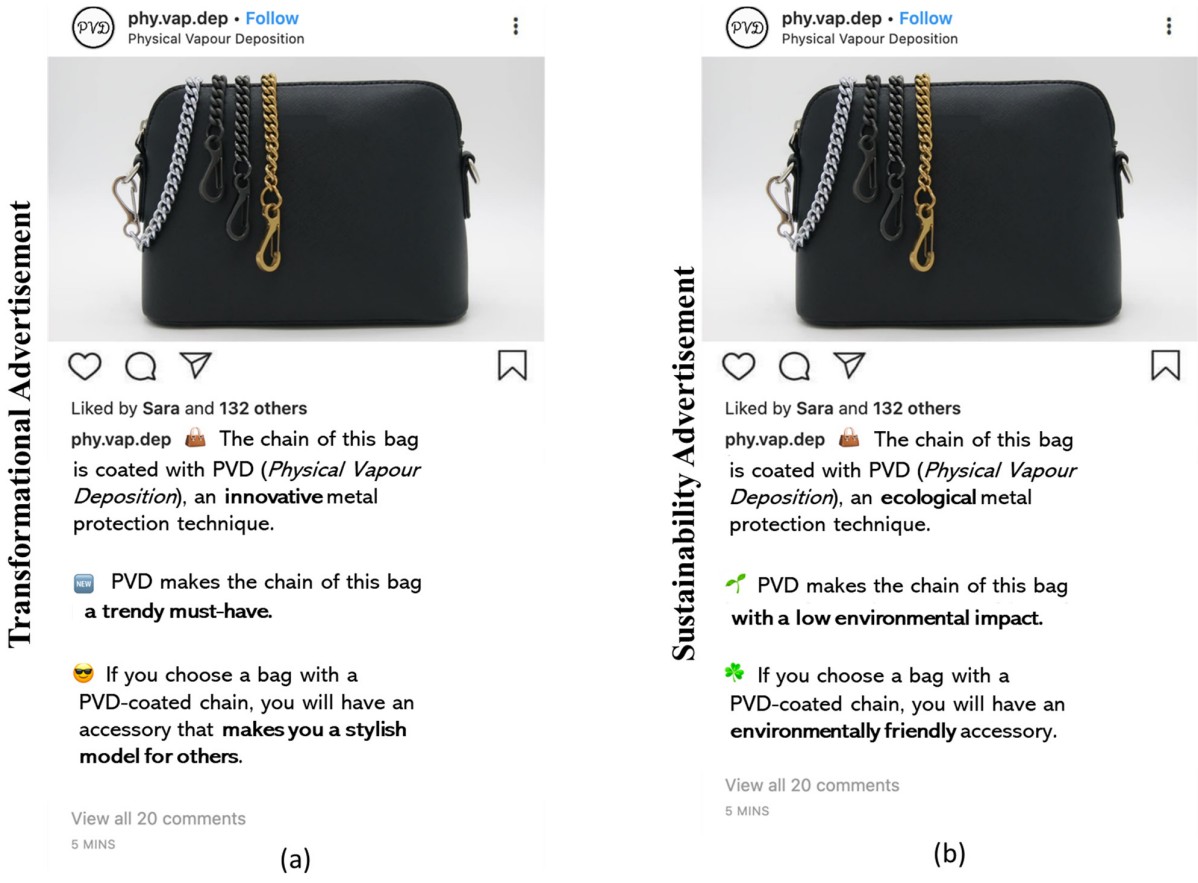

**Figure 2.** Instagram posts in the transformational (**a**) and sustainability (**b**) conditions.

Immediately after viewing the Instagram post, all participants replied to a series of questions about the advertisement and the bag. Those completing the entire research were paid £5.00 through the Prolific platform. A total of 517 participants accessed the questionnaire and completed it correctly. A total of 7 participants dropped out and were excluded from the final sample (N = 510; informational message: *n* = 129; transformational message: *n* = 129; sustainability message: *n* = 129; control message: *n* = 130; age range: 18–70 years; M = 28.69; SD = 9.53)

*3.2. Measures*

The questionnaire included several measures. Only the measures relevant to the present study are reported below.

The perfectionist consumer profile was assessed using four items on a 7-point Likert scale (e.g., "Getting very good quality is very important to me… completely disagree" (1) to "completely agree" (7); adapted from [49]; $\alpha$ = 0.80).

The novelty-fashion-seeking consumer profile was measured using four items on a 7-point Likert scale (e.g., "I usually have one or more trendy clothes… completely disagree (1)–completely agree (7)"; adapted from [49]; $\alpha$ = 0.82).

The hedonic consumer profile was assessed using three items on a 7-point Likert scale (e.g., "Shopping is a pleasant activity to me… completely disagree (1)–completely agree (7)"; adapted from [49]; $\alpha$ = 0.85).

The green consumer profile was measured using three items on a 7-point Likert scale (e.g., "I buy sustainable accessories… completely disagree (1)–completely agree (7)"; adapted from [61]; $\alpha$ = 0.85).

Message involvement was measured with three items using a Likert scale ranging from "completely disagree" (1) to "completely agree" (7) (e.g., "The Instagram post involved me"; $\alpha = 0.87$) [62].

Systematic processing was measured with five items using a Likert scale ranging from "completely disagree" (1) to "completely agree" (7) (e.g., "While I was reading the messages, I thought about what actions I might take based on what I read"; $\alpha = 0.84$), adapted from [63].

Intention to buy the bag prototype was measured with a single item (i.e., "Do you intend to buy a bag like the one you saw in the post?") ranging from "not at all" (1) to "absolutely yes" (5).

Willingness to pay for the bag prototype was measured with a single item (i.e., "How much would you pay for a bag similar to the one you saw in the post?"). In this case, participants were free to indicate any price they were willing to pay, with no limit on spending.

Finally, we asked participants for sociodemographic information including age, level of education, marital status, demographic dimension of the municipality of residence, and monthly income.

### 3.3. Data Analysis

All analyses were conducted with SPSS (version 25, SPSS Statistics/IBM Corp, Chicago, IL, USA). As preliminary analyses, we first ran descriptive and correlation analyses to explore the measured variables and the relationships among them. We then checked for the absence of biases in randomization and dropouts using chi-square tests and logistic regression. As to the main analyses, we ran a MANOVA to compare differently framed messages according to their capacity to trigger involvement, credibility, and systematic processing (RQ1). Then, we used MANOVA to investigate which message appeal was more effective in influencing consumers' intention to buy and willingness to pay for a bag prototype with the PVD-coated chain, compared to the control message (RQ2). Finally, to test our H1-H4, we ran a mediation analysis using Model 7 of the PROCESS macro for SPSS [64]. In this analysis, we considered the results with bootstrapped 95% confidence intervals (CI) that did not include zero to be significant (i.e., if a 95% confidence interval includes zero, then there is no statistically significant effect; if the confidence interval does not include zero, then there is a statistically significant effect).

### 4. Results

#### 4.1. Preliminary Analysis

As shown in Table 1, most participants were single, young, or young adults, with a high educational level, and with a monthly income below €1200. Table 2 reports the means, standard deviations, and correlations between the study variables. Table 3 shows the means and standard deviations of the study variables in each condition. Chi-square did not show any significant differences in age, level of education, marital status, demographic dimension of the municipality of residence, and monthly income across conditions (all $p > 0.10$). This suggests that randomization was adequate, with the four conditions being comparable as to the demographic characteristics of the participants. Regarding dropouts, seven participants dropped out. Chi-square did not show any significant differences in dropouts ($p = 0.67$) across conditions. In addition, the results of logistic regression showed that dropouts did not differ based on consumer decision-making style (all $p > 0.06$). Given the low rate of dropout and the absence of differences according to consumer decision-making styles, we can assert that the final sample was representative of the initial sample. All the analyses were conducted on the final sample.

**Table 1.** Demographics of the study sample.

|  | % Total Sample |
|---|---|
| Age | |
|     18–27 | 62.9 |
|     28–37 | 22.6 |
|     38–47 | 7.7 |
|     48–57 | 4.8 |
|     58–70 | 1.9 |
| Marital Status | |
|     Single | 67.7 |
|     Cohabiting Couple | 16.4 |
|     Married | 10.3 |
|     Separated/Divorced | 1.9 |
|     Widow | 0.4 |
| Education | |
|     Primary School | 0 |
|     Secondary School | 1 |
|     High School without Diploma | 1.5 |
|     High School Diploma | 16.1 |
|     University without Degree | 27.3 |
|     University Degree | 27.7 |
| Monthly Income | |
|     Low (up to 1200€) | 53.4 |
|     Medium (1200–1800€) | 13.9 |
|     High (more than 1801€) | 9.9 |
|     Not Declared | 22.8 |
| Number of Residents in the Municipality of Residence | |
|     Less than 10,000 | 16.1 |
|     Between 10,000 and 30,000 | 23.2 |
|     Between 30,000 and 100,000 | 23.4 |
|     Between 100,000 and 250,000 | 10.1 |
|     Between 250,000 and 500,000 | 3.9 |
|     More than 500,000 | 23.4 |

**Table 2.** Correlations among the study variables.

|  | **1.** | **2.** | **3.** | **4.** | *M* | *SD* |
|---|---|---|---|---|---|---|
| 1. Message Involvement | 1 | | | | 4.01 | 1.35 |
| 2. Systematic Processing | 0.64 ** | 1 | | | 3.88 | 1.25 |
| 3. Intention | 0.59 ** | 0.43 ** | 1 | | 2.58 | 1.00 |
| 4. Willingness to Pay | 0.31 ** | 0.22 ** | 0.32 ** | 1 | 40.09 | 35.30 |

*Note. M* = mean; *SD* = standard deviations. The numbers below diagonal are the correlation coefficients among the study variables. ** $p < 0.001$.

**Table 3.** Means and standard deviations of the study variables in each advertisement condition.

| | Control Advertisement Condition (*n* = 129) | | Informational Advertisement Condition (*n* = 129) | | Emotional Advertisement Condition (*n* = 129) | | Sustainability Advertisement Condition (*n* = 130) | |
|---|---|---|---|---|---|---|---|---|
| | *M* | *SD* | *M* | *SD* | *M* | *SD* | *M* | *SD* |
| Message Involvement | 3.63 | 1.25 | 4.09 | 1.35 | 3.60 | 1.34 | 4.73 | 1.15 |
| Systematic Processing | 3.60 | 1.26 | 3.88 | 1.25 | 3.71 | 1.25 | 4.32 | 1.12 |
| Intention to Buy | 2.60 | 0.86 | 2.55 | 1.14 | 2.33 | 0.99 | 2.82 | 0.95 |
| Willingness to Pay | 35.24 | 23.92 | 39.61 | 34.35 | 44.66 | 51.26 | 40.83 | 24.37 |

*Note. M* = mean, *SD* = standard deviation.

### 4.2. Message Evaluation

To answer our RQ1, we checked if there were differences in the message evaluation in the four message conditions. We used multivariate tests, which allowed us to compare the average responses of the participants of each condition in terms of message involvement (i.e., to what degree the message was involving) and systematic processing (i.e., to what extent the message was carefully elaborated). Results of the multivariate tests showed a significant main effect of the experimental condition ($F(9,1539) = 10.03$, $p = 0.001$, $\eta p2 = 0.05$). A univariate test confirmed the effect of the condition on message involvement and systematic processing (all $p < 0.001$), with participants exposed to different messages reporting different levels of message involvement and systematic processing. Specifically, Tukey's HSD test (i.e., a test assessing the significance of differences between pairs of condition means) highlighted that the sustainability message was the most involving one, followed by the informational message, which was more involving compared to the transformational, and the control message (Table 4). Finally, the sustainability message was the most systematically processed compared to the other messages. No other differences emerged.

**Table 4.** Mean differences in study variables in each experimental condition.

| Dependent Variable | Condition | Condition | Mean Difference |
|---|---|---|---|
| Message involvement | Informational Advertisement | Control Advertisement | 0.46 * |
| | | Emotional Advertisement | 0.48 * |
| | | Sustainability Advertisement | −0.64 ** |
| | Emotional Advertisement | Control Advertisement | −0.02 |
| | | Informational Advertisement | −0.48 * |
| | | Sustainability Advertisement | −1.13 ** |
| | Sustainability Advertisement | Control Advertisement | 1.10 ** |
| | | Informational Advertisement | 0.64 ** |
| | | Emotional Advertisement | 1.13 ** |
| Systematic Processing | Informational Advertisement | Control Advertisement | 0.28 |
| | | Emotional Advertisement | 0.17 |
| | | Sustainability Advertisement | −0.44 * |
| | Emotional Advertisement | Control Advertisement | 0.11 |
| | | Informational Advertisement | −0.17 |
| | | Sustainability Advertisement | −0.61 ** |
| | Sustainability Advertisement | Control Advertisement | 0.72 ** |
| | | Informational Advertisement | 0.44 * |
| | | Emotional Advertisement | 0.61 ** |
| Intention to Buy | Informational Advertisement | Control Advertisement | −0.05 |
| | | Emotional Advertisement | 0.22 |

| | | Sustainability Advertisement | −0.27 |
| | Emotional Advertisement | Control Advertisement | −0.28 |
| | | Informational Advertisement | −0.22 |
| | | Sustainability Advertisement | −0.50 ** |
| | Sustainability Advertisement | Control Advertisement | 0.22 |
| | | Informational Advertisement | 0.27 |
| | | Emotional Advertisement | 0.50 ** |
| Willingness to Pay | Informational Advertisement | Control Advertisement | 4.37 |
| | | Emotional Advertisement | −5.05 |
| | | Sustainability Advertisement | −1.22 |
| | Emotional Advertisement | Control Advertisement | 9.42 |
| | | Informational Advertisement | 5.05 |
| | | Sustainability Advertisement | 3.83 |
| | Sustainability Advertisement | Control Advertisement | 5.59 |
| | | Informational Advertisement | 1.22 |
| | | Emotional Advertisement | −3.83 |

*Note.* \* $p < 0.05$, \*\* $p < 0.001$.

### 4.3. Effects of Messages on Intention and Willingness to Buy the Bag Prototype

To answer our RQ2 on the most effective message in influencing consumers' intention to buy and willingness to pay for a bag prototype with a PVD-coated chain, we ran a MANOVA. Results of the multivariate tests showed a significant main effect of condition ($F_{(6,1026)} = 4.42$, $p = 0.001$, $\eta p2 = 0.02$). Overall, participants were persuaded differently by messages based on the condition to which they had been assigned. Univariate tests revealed a main significant effect of condition on the participants' intention to buy the bag prototype ($F_{(3,513)} = 5.50$, $p = 0.001$, $\eta p2 = 0.09$), but not on their willingness to pay for it ($F_{(3,513)} = 1.56$, $p = 0.20$, $\eta p2 = 0.09$). Tukey's HSD test (Table 4) highlighted that participants who were in the sustainability condition had a higher intention to buy the bag as compared to those who were in the transformational condition. No other differences among conditions emerged.

### 4.4. Effects of Messages on Intention as a Function of Participants' Decision-Making Style

To test our H1–H4, we ran a series of moderated mediation analyses (Model 7 of the PROCESS macro for SPSS; [64]). These analyses allowed us to verify whether the involvement in the messages and their systematic processing influenced the intentions to buy the bag as a function of consumers' decision-making styles. In the first series of analyses, we tested whether the interaction between condition and consumers' decision-making style influenced message involvement and systematic processing, which, in turn, would mediate the relationship between the condition and the intention to buy the bag prototype. In a second series of analyses, we tested the same paths of interactions, but this time with willingness to pay for the bag prototype as the dependent variable. Below, we discuss only the moderated mediation analyses showing a significant interaction between consumers' decision-making style and message conditions.

#### 4.4.1. Perfectionist Consumers

When exposed to the informational message, participants with medium and high levels of perfectionism intended to buy the bag more than those with low levels of perfectionism ($B = 0.23$; *95% CI* = 0.01, 0.47; Figure 3a) and were willing to pay a higher price for it ($B = 8.76$; *95% CI* = 0.41, 17.10; Figure 3b). They were also more involved in the messages and, in turn, had higher intention to buy (*Ind. Effect* = 0.14; *95% CI* = 0.03, 0.26; Table 5) and willingness to pay for the bag (*Ind. Effect* = 3.40; *95% CI* = 0.72, 6.93; Table 5). Therefore, we fully confirmed our H1a–c. In the transformational message condition, the

participants' level of perfectionism did not influence the effect of message involvement and systematic processing on the intention to buy the bag and willingness to pay for it. In the sustainability message condition, the more perfectionist consumers were, the more they perceived the message as involving, and in turn intended to buy the bag (*Ind. Effect* = 0.12; *95% CI* = 0.01, 0.22; Table 5) and pay more for it (*Ind. Effect* = 2.72; *95% CI* = 0.34, 5.65; Table 5). Therefore, we also fully confirmed Hd–f. All results of these moderated mediation analyses are reported in Table 5.

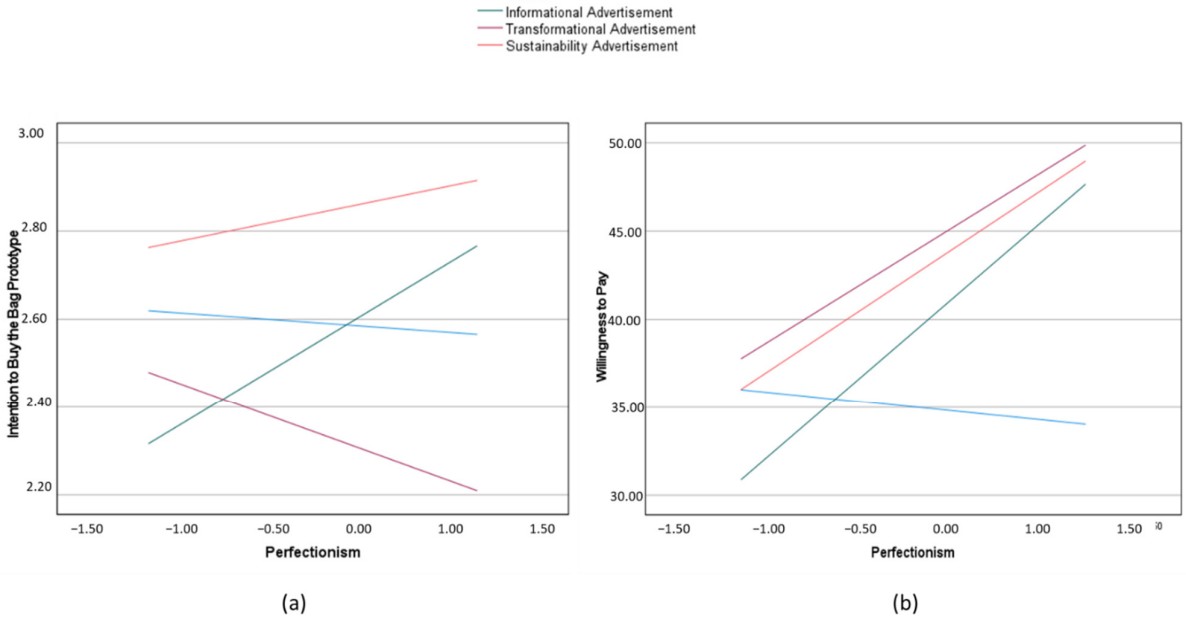

**Figure 3.** Consumers' intention (a) and willingness to pay (b) for the bag prototype at different levels of perfectionism.

**Table 5.** Message effects according to participant's levels of perfectionism.

| | Relative Conditional Indirect Effects of Conditions on Intention to Buy | | | Indexes of the Moderated Mediations | Relative Conditional Indirect Effects of Conditions on Willingness to Pay | | | Indexes of the Moderated Mediations |
|---|---|---|---|---|---|---|---|---|
| | Low Perfectionism | Medium Perfectionism | High Perfectionism | | Low Perfectionism | Medium Perfectionism | High Perfectionism | |
| **Informational Advertisement Condition** | | | | | | | | |
| Message Involvement | −0.01 (95% CI = −0.16, 0.14) | 0.13 * (95% CI = 0.03, 0.24) | 0.30 * (95% CI = 0.13, 0.50) | 0.14 * (95% CI = 0.03, 0.26) | −0.14 (95% CI = −4.00, 3.41) | 3.26 * (95% CI = 0.79, 6.35) | 2.71 * (95% CI = 2.91, 13.58) | 3.40 * (95% CI = 0.72, 6.93) |
| Systematic Processing | 0.00 (95% CI = −0.03, 0.04) | 0.02 (95% CI = −0.00, 0.05) | 0.03 (95% CI = −0.00, 0.10) | 0.01 (95% CI = −0.01, 0.05) | 0.02 (95% CI = −0.78, 0.88) | 0.15 (95% CI = −0.67, 1.08) | 0.31 (95% CI = −1.37, 2.04) | 0.13 (95% CI = −0.64, 1.05) |
| **Emotional Advertisement Condition** | | | | | | | | |
| Message Involvement | −0.05 (95% CI = −0.20, 0.11) | −0.01 (95% CI = −0.11, 0.09) | 0.04 (95% CI = −0.11, 0.20) | 0.04 (95% CI = −0.06, 0.15) | −0.12 (95% CI = −5.61, 2.56) | −0.22 (95% CI = −2.80, 2.26) | 1.04 (95% CI = −2.64, 5.22) | 1.01 (95% CI = −1.55, 4.05) |
| Systematic Processing | 0.00 (95% CI = −0.04, 0.04) | 0.00 (95% CI = −0.02, 0.03) | 0.01 (95% CI = −0.02, 0.06) | 0.00 (95% CI = −0.02, 0.03) | 0.00 (95% CI = −0.71, 0.81) | 0.05 (95% CI = −0.45, 0.62) | 0.11 (95% CI = −0.85, 1.08) | 0.05 (95% CI = −0.55, 0.63) |
| **Sustainability Advertisement Condition** | | | | | | | | |
| Message Involvement | 0.24* (95% CI = 0.11, 0.39) | 0.34* (95% CI = 0.23, 0.48) | 0.49 * (95% CI = 0.30, 0.70) | 0.12 * (95% CI = 0.01, 0.22) | 5.88 * (95% CI = 1.41, 10.59) | 8.61 * (95% CI = 4.93, 13.29) | 12.02 * (95% CI = 6.51, 18.90) | 2.72 * (95% CI = 0.34, 5.65) |
| Systematic Processing | 0.04 (95% CI = −0.00, 0.11) | 0.05 (95% CI = −0.00, 0.11) | 0.06 (95% CI = −0.01, 0.14) | 0.01 (95% CI = −0.02, 0.03) | 0.37 (95% CI = −1.43, 2.39) | 0.43 (95% CI = −1.60, 2.55) | 0.52 (95% CI = −2.00, 3.19) | 0.07 (95% CI = −0.50, 0.75) |

*Note.* * Significant conditional effects.

### 4.4.2. Novelty-Fashion-Seeking Consumer

As in the case of perfectionist consumers, the more novelty-fashion-seeking consumers were, the more they intended to buy the bag when exposed to the informational advertisement ($B$ = 0.21; *95% CI* = 0.01, 0.33; Figure 4). In addition, the more they perceived the informational advertisement as involving, the more they intended to buy the bag (*Ind. Effect* = 0.08; *95% CI* = 0.01, 0.16; Table 6) and wanted to pay for the bag (*Ind. Effect* = 2.11; *95% CI* = 0.36, 4.28; Table 6). However, the persuasiveness of the transformational and sustainability advertisements did not vary according to the consumers' level of novelty-fashion consciousness. Therefore, we did not confirm our H2.

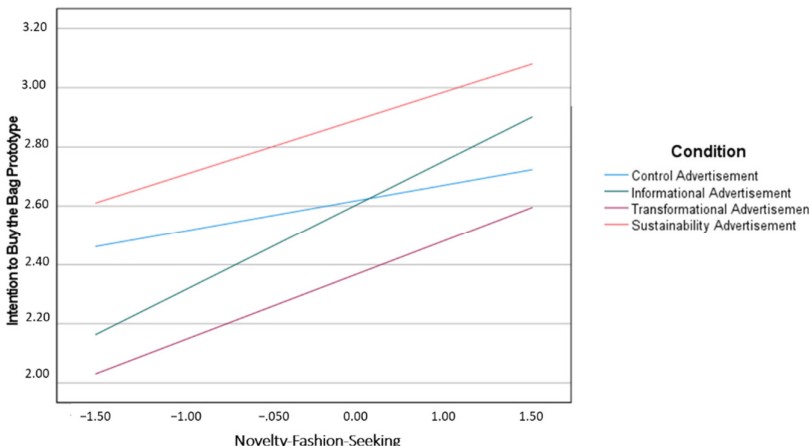

**Figure 4.** Consumers' intention and willingness to pay for the bag prototype at different levels of novelty-fashion seeking.

**Table 6.** Message effects according to participant's levels of novelty-fashion consciousness.

| | Relative Conditional Indirect Effects of Conditions on Intention to Buy | | | Indexes of the Moderated Mediations | Relative Conditional Indirect Effects of Conditions on Willingness to Pay | | | Indexes of the Moderated Mediations |
|---|---|---|---|---|---|---|---|---|
| | Low Novelty-Fashion Consciousness | Medium Novelty-Fashion Consciousness | High Novelty-Fashion Consciousness | | Low Novelty-Fashion Consciousness | Medium Novelty-Fashion Consciousness | High Novelty-Fashion Consciousness | |
| **Informational Advertisement Condition** | | | | | | | | |
| Message Involvement | 0.01 (*95% CI =* −0.11, 0.14) | 0.12 * (*95% CI =* 0.03, 0.23) | 0.27 * (*95% CI =* 0.10, 0.45) | 0.08 * (*95% CI =* 0.01, 0.16) | 0.24 (*95% CI =* −2.86, 3.79) | 3.06 * (*95% CI =* 0.76, 6.12) | 6.59 * (*95% CI =* 2.55, 11.88) | 2.11 * (*95% CI =* 0.36, 4.28) |
| Systematic Processing | 0.01 (*95% CI =* −0.03, 0.04) | 0.02 (*95% CI =* −0.00, 0.05) | 0.03 (*95% CI =* −0.01, 0.09) | 0.01 (*95% CI =* −0.01, 0.03) | 0.06 (*95% CI =* −0.72, 0.89) | 0.14 (*95% CI =* −0.64, 1.09) | 0.25 (*95% CI =* −1.04, 1.91) | 0.06 (*95% CI =* −0.37, 0.68) |
| **Emotional Advertisement Condition** | | | | | | | | |
| Message Involvement | −0.11 (*95% CI =* −0.24, 0.01) | −0.02 (*95% CI =* −0.11, 0.07) | 0.10 (*95% CI =* −0.02, 0.23) | 0.07 (*95% CI =* −0.00, 0.14) | −2.82 (*95% CI =* −6.49, 0.53) | −0.51 (*95% CI =* −2.98, 1.80) | 2.38 (*95% CI =* −1.29, 6.59) | 1.73 * (*95% CI =* 0.01, 3.70) |
| Systematic Processing | 0.01 (*95% CI =* −0.03, 0.05) | 0.01 (*95% CI =* −0.02, 0.03) | 0.01 (*95% CI =* −0.03, 0.05) | 0.00 (*95% CI =* −0.02, 0.03) | 0.00 (*95% CI =* −2.10, 3.49) | 0.06 (*95% CI =* −1.60, 2.53) | 0.06 (*95% CI =* −1.09, 1.78) | 0.00 (*95% CI =* −0.43, 0.50) |
| **Sustainability Advertisement Condition** | | | | | | | | |
| Message Involvement | 0.33 * (*95% CI =* 0.07, 0.48) | 0.34 * (*95% CI =* 0.23, 0.47) | 0.36 * (*95% CI =* 0.19, 0.55) | 0.01 (*95% CI =* −0.06, 0.08) | 8.11 * (*95% CI =* 4.21, 13.57) | 8.23 * (*95% CI =* 4.71, 13.17) | 8.83 * (*95% CI =* 4.20, 14.57) | 0.24 (*95% CI =* −1.47, 2.04) |
| Systematic Processing | 0.06 (*95% CI =* −0.01, 0.15) | 0.05 (*95% CI =* −0.00, 0.11) | 0.03 (*95% CI =* −0.01, 0.08) | −0.01 (*95% CI =* −0.04, 0.00) | 0.56 (*95% CI =* −2.10, 3.49) | 0.42 (*95% CI =* −1.60, 2.54) | 0.23 (*95% CI =* −2.10, 3.49) | −0.11 (*95% CI =* −0.86, 0.50) |

Note. * Significant conditional indirect effects.

### 4.4.3. Hedonic Consumers

The effectiveness of the transformational advertisement varied as a function of hedonic decision-making style ($B$ = 0.20; *95% CI* = 0.04, 0.37). That is, consumers with a low level of hedonism had a lower intention to buy the bag when exposed to the transformational advertisement compared to the other advertisements. However, consumers with a high level of hedonism did not prefer the transformational advertisement; thus, our H3 was not supported (Figure 5). Concerning participants in the informational advertisement condition, the more hedonic they were, the more they perceived the informational advertisement as involving and, in turn, intended to buy and pay for the bag more. Regarding the transformational and sustainability advertisements, message involvement and systematic processing did not differently impact consumers' intention and willingness to pay as a function of the consumers' level of hedonism. All results of these moderated mediation analyses are reported in Table 7.

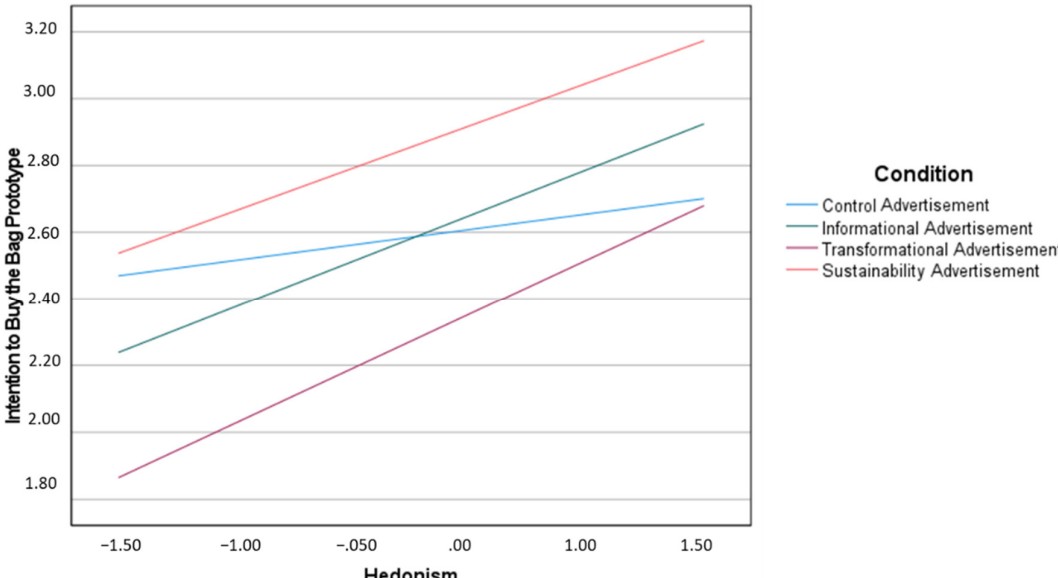

**Figure 5.** Consumers' intention and willingness to pay for the bag prototype at different levels of hedonism.

**Table 7.** Message effects according to participant's levels of hedonism.

| | Relative Conditional Indirect Effects of Conditions on Intention to Buy | | | Indexes of the Moderated Mediations | Relative Conditional Indirect Effects of Conditions on Willingness to Pay | | | Indexes of the Moderated Mediations |
|---|---|---|---|---|---|---|---|---|
| | Low Hedonism | Medium Hedonism | High Hedonism | | Low Hedonism | Medium Hedonism | High Hedonism | |
| **Informational Advertisement Condition** | | | | | | | | |
| Message Involvement | 0.05 (*95% CI* = −0.10, 0.20) | 0.17 * (*95% CI* = 0.07, 0.28) | 0.25 * (*95% CI* = 0.10, 0.43) | 0.05 * (*95% CI* = 0.01, 0.15) | 1.28 (*95% CI* = −2.43, 5.27) | 4.20 * (*95% CI* = 1.59, 7.53) | 6.14 * (*95% CI* = 2.13, 11.25) | 1.46 (*95% CI* = −0.24, 3.45) |
| Systematic Processing | 0.02 (*95% CI* = −0.01, 0.07) | 0.02 (*95% CI* = −0.01, 0.07) | 0.02 (*95% CI* = −0.00, 0.06) | 0.00 (*95% CI* = −0.02, 0.02) | 0.19 (*95% CI* = −0.98, 1.49) | 0.19 (*95% CI* = −0.75, 1.26) | 0.19 (*95% CI* = −0.79, 1.54) | −0.00 (*95% CI* = −0.34, 0.42) |
| **Emotional Advertisement Condition** | | | | | | | | |
| Message Involvement | −0.14 (*95% CI* = −0.32, 0.01) | −0.01 (*95% CI* = −0.11, 0.09) | 0.08 (*95% CI* = −0.06, 0.23) | 0.07 (*95% CI* = −0.00, 0.14) | −3.49 (*95% CI* = −8.20, 0.62) | −0.24 (*95% CI* = −2.63, 2.10) | 1.98 (*95% CI* = −1.64, 5.85) | 1.62 (*95% CI* = −0.23, 3.71) |
| Systematic Processing | 0.08 (*95% CI* = −0.01, 0.17) | 0.04 (*95% CI* = −0.00, 0.10) | 0.02 (*95% CI* = −0.01, 0.07) | −0.01 (*95% CI* = −0.04, 0.00) | 0.14 (*95% CI* = −0.90, 1.40) | 0.05 (*95% CI* = −0.43, 0.61) | −0.01 (*95% CI* = −0.80, 0.74) | −0.04 (*95% CI* = −0.58, 0.39) |
| **Sustainability Advertisement Condition** | | | | | | | | |
| Message Involvement | 0.35 * (*95% CI* = 0.19, 0.52) | 0.35 * (*95% CI* = 0.23, 0.48) | 0.35 * (*95% CI* = 0.19, 0.53) | 0.00 (*95% CI* = −0.06, 0.07) | 8.58 * (*95% CI* = 4.09, 14.32) | 8.58 * (*95% CI* = 4.63, 13.21) | 8.58 * (*95% CI* = 4.33, 13.91) | −0.00 (*95% CI* = −1.75, 1.70) |
| Systematic Processing | 0.08 (*95% CI* = −0.01, 0.10) | 0.04 (*95% CI* = −0.00, 0.10) | 0.02 (*95% CI* = −0.00, 0.07) | −0.02 (*95% CI* = −0.04, 0.00) | 0.67 (*95% CI* = −2.42 3.96) | 0.39 (*95% CI* = −1.40, 2.29) | 0.20 (*95% CI* = −0.86, 1.4) | −0.14 (*95% CI* = −1.00, 0.55) |

Note. * Significant conditional indirect effects.

### 4.4.4. Green Consumers

Participants' level of green consumerism did not interact with any advertisement condition. Therefore, the sustainability message was the most persuasive message, regardless of people's attention to environmental protection (as we had hypothesized with H4a). However, the more participants were green consumers, the more they found sustainability advertisements involving (*Ind. Effect* = 0.09; *95% CI* = 0.01, 0.19; Table 8) and the more they processed these advertisements carefully (*Ind. Effect* = 0.01; *95% CI* = 0.01, 0.14; Table 8). Therefore, we confirmed our H4b and H4c. Finally, green participants' willingness to pay did not differ across advertisement conditions.

**Table 8.** Message effects according to participant's levels of green consumerism.

| | Relative Conditional Indirect Effects of Conditions on Intention to Buy | | | Indexes of the Moderated Mediations | Relative Conditional Indirect Effects of Conditions on Willingness to Pay | | | Indexes of the Moderated Mediations |
|---|---|---|---|---|---|---|---|---|
| | Low Green Consumerism | Medium Green Consumerism | High Green Consumerism | Moderated Mediations | Low Green Consumerism | Medium Green Consumerism | High Green Consumer | |
| **Informational Message Condition** | | | | | | | | |
| Message Involvement | 0.09 (*95% CI* = −0.06, 0.26) | 0.13 * (*95% CI* = 0.03, 0.25) | 0.17 * (*95% CI* = 0.05; 0.31) | 0.04 (*95% CI* = −0.05, 0.13) | 2.01 (*95% CI* = −2.73, 6.86) | 3.61 * (*95% CI* = 0.59, 7.12) | 5.21 * (*95% CI* = 1.43, 9.99) | 1.03 (*95% CI* = −1.30, 3.46) |
| Systematic Processing | 0.02 (*95% CI* = −0.01, 0.08) | 0.02 (*95% CI* = −0.00, 0.06) | 0.02 (*95% CI* = −0.01, 0.06) | −0.00 (*95% CI* = −0.03, 0.02) | 0.19 (*95% CI* = −0.93, 1.65) | 0.16 (*95% CI* = −0.74, 1.22) | 0.14 (*95% CI* = −0.75, 1.08) | −0.02 (*95% CI* = −0.63, 0.39) |
| **Emotional Message Condition** | | | | | | | | |
| Message Involvement | −0.08 (*95% CI* = −0.21, 0.06) | −0.06 (*95% CI* = −0.10, 0.10) | 0.06 (*95% CI* = −0.06, 0.20) | 0.07 (*95% CI* = −0.02, 0.16) | −1.87 (*95% CI* = −5.64, 1.53) | −0.16 (*95% CI* = −2.77, 2.29) | 1.60 (*95% CI* = −1.59, 5.17) | 1.70 (*95% CI* = −0.41, 4.16) |
| Systematic Processing | 0.00 (*95% CI* = −0.03, 0.04) | 0.01 (*95% CI* = −0.01, 0.04) | 0.02 (*95% CI* = −0.01, 0.06) | 0.01 (*95% CI* = −0.01, 0.04) | 0.03 (*95% CI* = −0.60, 0.88) | 0.10 (*95% CI* = −0.54, 0.92) | 0.17 (*95% CI* = −0.89, 1.34) | 0.07 (*95% CI* = −0.51, 0.67) |
| **Sustainability Message Condition** | | | | | | | | |
| Message Involvement | 0.24 * (*95% CI* = 0.11, 0.41) | 0.34 * (*95% CI* = 0.23, 0.47) | 0.43 * (*95% CI* = 0.28, 0.61) | 0.09 * (*95% CI* = 0.01, 0.19) | 6.05 * (*95% CI* = 2.45, 10.67) | 8.34 * (*95% CI* = 4.66, 12.90) | 10.72 * (*95% CI* = 5.71, 16.80) | 2.29 (*95% CI* = −0.15, 5.06) |
| Systematic Processing | 0.04 * (*95% CI* = 0.01, 0.11) | 0.05 * (*95% CI* = 0.01, 0.11) | 0.06 * (*95% CI* = 0.01, 0.14) | 0.01 * (*95% CI* = 0.01, 0.04) | 0.33 (*95% CI* = −1.25, 2.36) | 0.44 (*95% CI* = −1.61, 2.66) | 0.56 (*95% CI* = −2.08, 3.26) | 0.11 (*95% CI* = −0.58, 0.78) |

Note. * Significant conditional indirect effects.

## 5. Discussion

In the present study, we tested how to convince people to buy an innovative and sustainable bag by levering informational, transformational, or sustainability appeals in online advertisements. The study analyzed how these advertisements are evaluated and influence consumers' intention to buy a bag coated with PVD (physical vapor deposition) and their willingness to pay for the bag. Moreover, we explored whether the effects of each appeal depended on the decision-making style of the consumer exposed to it.

We found that the exposure to a fictional Instagram post with a sustainability appeal, that is, an advertisement informing about the low environmental impact of the bag, was perceived as more involving and was more systematically processed than the posts with informational or transformational appeals. Notably, participants exposed to the sustainability appeal intended to buy the bag more than participants exposed to the other appeals. This finding extends what has already been found in studies related to the introduction of sustainable innovations in markets other than the fashion market (e.g., food market; [59,60]). This result also suggests that, when introducing an accessory produced with an innovative technology, it is more effective to refer explicitly to its sustainability benefits (in our example, referring directly to the reduced environmental impact of PVD technology) rather than focus on other functional benefits, such as durability and endurance. This is in line with research suggesting that the effectiveness of explicit messages may be influenced by factors such as product complexity and product involvement.

Given that apparel is not a complex product and that many consumers are highly involved in the purchase of apparel owing to its symbolic and hedonic characteristics [5,38], fashion brands typically rely on implicit messages, often brand name only, to communicate meanings and values to consumers. However, when the fashion product

presents innovations that are complex to explain to consumers, such explicit communication may be desirable, because it enhances involvement and activates the systematic processing of the advertisement content.

An alternative explanation of the superior effectiveness of a post on the sustainable impact of innovative technology might be that people want to buy sustainable products to communicate important aspects of their identity and social responsibility. Buying behaviors, including those related to innovative products, are used to symbolize people's values, identities, and memberships [65,66]. We could therefore hypothesize that our sample of young Italian women was interested in making purchases that express their attention and concern towards environmental issues.

Concerning the effectiveness of each advertisement as a function of the consumer's decision-making style, first we found that perfectionism activates a higher involvement in and systematic processing of informational messages, that is, advertisements focused on the intrinsic characteristics of the product. For perfectionist consumers, the features of "durability" and "resistance" were likely synonymous with quality and therefore attracted their attention and raised their desire to purchase. The same consumers, however, were also attracted by messages related to the environmental impact of the fashion production. This result denotes how fashion sustainability is increasingly perceived as a salient attribute connected to the perception of high product quality, as past studies have already demonstrated in the case of other consumers' choices [67].

As with the perfectionist consumers, the novelty-fashion-seeking consumer was also very convinced by the informational advertisement. This result contrasts with what was hypothesized and sheds new light on this category of consumers. We expected that keywords such as "innovative product" would especially attract the attention of this type of consumer. Instead, we discovered that the proposal of a new technology that increases the functional feature of the product (i.e., resistance and durability) was more convincing for them. Probably, these consumers are characterized by greater attention to novelties that denote a greater performance of the products rather than to novelty per se.

As to the hedonic decision-making style, we found that a low level of hedonism reduced the impact of a transformational appeal. However, we did not conversely find that a high level of hedonism increased the persuasiveness of the transformational appeal. Instead, we observed that hedonic consumers were more involved and, in turn, intended to buy and pay more for the bag when exposed to the informational appeal. This suggests that consumers, who buy clothing for pleasure and positive emotions, are also affected by the functional aspects of the products. Furthermore, we cannot rule out that the hedonists' lack of interest in our transformational advertising might have been due to its content or presentation. It is probable that our message, which particularly focused on the emotions that would derive from being a fashion futurist, did not adequately match the peculiar characteristics of this group.

A final interesting result of this study concerns the fact that green consumers are attracted only by the sustainability appeal and not by the informational one, although it implicitly referred to a benefit in terms of sustainability. As a matter of fact, increasing resistance and durability is a means of reducing the consumption of environmental resources to produce new accessories. However, green consumers seem to be more persuaded by the explicit message on the sustainability of the product. If the intent is to involve and persuade green consumers to buy a sustainable and new product, using simple and clear messages on why the product is sustainable seems to be more effective than giving more complex explanations on the reasons underlying the sustainability of the new production method.

To sum up, this study offers several contributions to the development of advertisement related to sustainable fashion produced using new technology. Notably, this study emphasizes the presence of a new trend in fashion purchasing, which is characterized by relevant attention to pro-environmental solutions in the fashion industry. In addition, even if we observed that the sustainability appeal is the best solution to promote this type

of innovation, we also demonstrated that the use of a rational appeal can be effective. Referring to the functional benefits of using new technologies to produce accessories is an effective strategy to promote purchasing when consumers are very interested in the high quality of the product, when they engage in shopping for fun and enjoyment, or when they are interested in novelty.

*Limitations and Future Directions*

Although this study offers several insights on how to promote sustainable accessories in the social media context, it is not exempt from limitations. First, the sample was not representative of the general Italian population, because they were not fully balanced in terms of the main sociodemographic variables. In addition, our sample had a predominance of young, single, and highly educated females. Our data should therefore be generalized with caution, and future research could usefully test the predictivity of this model in other populations. For example, future studies might test the effectiveness of diverse appeals for advertisements of male bags or other accessories. We cannot exclude that what has been observed here also applies when the type of accessory or item of clothing advertised is different.

Second, the observed effects of the interaction between consumers' decision-making styles and message conditions on willingness to pay had a very wide confidence interval. This may be attributed to the choice of a continuous measurement in which participants were asked to indicate any price they were willing to pay to purchase the bag. This type of measurement determined the collection of highly variable data distributed over a wide range of values. Indeed, there was considerable individual variation in the consumers' willingness to pay for the bag, as also shown by the wide standard deviations within message groups. Our findings should therefore be interpreted with caution, and future research might overcome this limitation by opting for a more specific measure of the consumers' willingness to pay, establishing an a priori range of the selectable prices.

Third, in our studies we did not control for the role of other individual characteristics (e.g., mavenism or political orientation [68]). Fourth, although the choice of accessories may be driven by different functional motivations, the informational appeal used in our study was only related to resistance and durability. Similarly, the transformational appeal was only based on a value-expressive motivation, and the sustainability appeal was only based on the pro-environmental benefits. Future studies could usefully compare different contents for each appeal (e.g., social corporate responsibility in the case of the sustainability appeal) to evaluate whether the effectiveness of such advertisements differs, also taking into consideration other individual differences among the receivers (e.g., self-interest vs. altruism). Finally, we collected data in a single time, limiting our analyses to intentions/willingness to pay. Future studies could consider adopting a longitudinal design, including behavioral measurements, and considering to what extent the observed intention and willingness to pay are translated into actual behavior.

As to the practical implications deriving from our findings, the present research offers diverse insights into public actions and campaigns aimed at promoting the purchasing of innovative sustainable fashion. First, we found that our advertisement with a simple sustainability appeal was effective in enhancing consumers' intention to buy the bag. This result can be usefully applied to future campaigns aimed at convincing women to adopt sustainable shopping, independent of their decision-making style.

Second, our findings showed that decision-making styles influence the elaboration and evaluation of the message content of diverse advertisement appeals as well as purchasing decisions. Interventions and communication should therefore be tailored accordingly to maximize the effectiveness in promoting sustainable fashion. It appears that a simple focus on the low environmental impact of the advertised product seems to be a promising approach with different kinds of consumers. It activates involvement and in-depth information processing in most of them, and these are translated into higher purchasing intention.

Third, our results offer some insights into how to employ psychological measures (such as the measures of consumers' decision-making styles) to better design digital marketing strategies (i.e., a set of techniques developed on the Internet to persuade users to buy a product or service [69]). The psychological measures employed in this study (i.e., consumers' decision-making styles) have been useful to identify online buyer persons, that is, archetypes of real buyers that might allow marketers to design personalized strategies to promote sustainable accessories. The definition of the buyer personas, and thus the identification of the audience segments, might also be extremely useful to develop customer-centric marketing [70], which can, in turn, be based on a programmatic advertising [71]. Pragmatic advertising is the automation and optimization of the purchase of advertising units to send the right message, at the right time, and to specific audience segments identified through careful targeting [72]. It represents one of the most interesting evolutions in the panorama of the online advertising market and can be particularly enriched by data collected through psychological measures. This could be achieved by integrating digital marketing and data sciences. With this integration, companies will be able to better manage the information collected from users and to more easily apply new data analyses and innovative techniques to create knowledge [73,74].

## 6. Conclusions

The present research contributes to our understanding of how to promote the diffusion of innovative technology in the sustainable fashion market online. Compared to the informational and transformational appeals, the sustainability appeal promoting a bag produced with a new technology with low environmental impact involves consumers more and, in turn, has a greater impact on their purchasing intentions.

Moreover, we demonstrated that consumers' willingness to pay for an innovative bag has a great individual variability and needs to be deeply investigated by referring to their decision-making styles. Indeed, consumers with different psychological features differ in their perception of message appeals as involving and in systematically processing them, and, therefore, in their higher or lower intention to buy the advertised sustainable product as well as to pay for it.

Perfectionist consumers are attracted by both informational and sustainability advertisements, especially when they perceive these messages as involving. Thus, the resistance and sustainability of the product attract the attention of these consumers and raise their desire to purchase it. The novelty-fashion-seeking consumers are only convinced by the informational advertisement. They are interested in purchasing an innovative accessory only if the new technology employed improves the rational product-related attributes. However, they are not attracted by messages proposing innovation as related to an aesthetic or pro-environment improvement. Hedonic consumers are only involved by the transformational advertisement. This finding probably reflects the fact that hedonic consumers perceive only high-quality bags as luxury and, therefore, connected to a high social status and a positive reputation. Finally, as expected, green consumers are attracted only by the sustainability appeal, because they perceive pro-environmental information as involving, and they carefully reflect on it.

Overall, these results enhance our understanding of how to use psychological measures to better define buyer personas and, in turn, craft advertising in line with their features.

**Author Contributions:** Conceptualization, V.C. and P.C.; methodology, V.C.; formal analysis, V.C.; investigation, V.C; data curation, V.C.; writing—original draft preparation, V.C. and P.C.; writing—review and editing, V.C. and P.C.; supervision, P.C.; project administration, P.C.; funding acquisition, P.C. All authors have read and agreed to the published version of the manuscript."

**Funding:** This research was funded by Regione Lombardia, grant number "Innovazione di processo e di prodotto nel settore della moda attraverso la eliminazione di materiali base e finiture con elevato impatto ambientale e loro sostituzione con soluzioni ecosostenibili" - (FriENdly.Pro) POR FESR - 2014-2020: ID 1311867 - Bando FASHIONTECH CUP E21B19000750007.

**Institutional Review Board Statement:** The study was conducted according to the guidelines of the Declaration of Helsinki, and approved by the Institutional Review Board (or Ethics Committee) of Università Cattolica del Sacro Cuore, Milano (protocol code 0618 and date of approval 4/04/2019).

**Informed Consent Statement:** Informed consent was obtained from all subjects involved in the study.

**Data Availability Statement:** The data presented in this study are available on request from the corresponding author.

**Conflicts of Interest:** The authors declare no conflict of interest. The funders had no role in the design of the study; in the collection, analyses, or interpretation of data; in the writing of the manuscript, or in the decision to publish the results.

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
