# Peer review of "Advertising Innovative Sustainable Fashion: Informational, Transformational, or Sustainability Appeal?"

_sustainability, doi:10.3390/su142316148_

Round 1
Reviewer 1 Report
This study focuses on the following question ; how to promote the diffusion of innovative technology in the sustainable fashion market ? This paper is well structured, well organized and well written.
This paper can be accepted pending the following revisions and justifications.
Ø Please, improve the quality of the figures. Figures and legends must be clearly readable.
Ø Authors should revise the citations of the references in the text. In fact, In the text, reference numbers should be placed in square brackets [ ], and placed before the punctuation; for example [1], [1–3] or [1,3].
Author Response
Dear Reviewer,
We thank you for the positive feedback on our paper. In revising the manuscript, we have addressed all the concerns that you provided for our manuscript. We have also enhanced the reference list to better support the theoretical background of the study. In addition, we have extensively revised the English language and style.
We hope the new version will be suitable for publication in Sustainability.
Please, find below a detailed response to your points, with the original comments in italics.
Thank you for your revision.
Best regards,
The Authors
***
This study focuses on the following question ; how to promote the diffusion of innovative technology in the sustainable fashion market ? This paper is well structured, well organized and well written.
This paper can be accepted pending the following revisions and justifications.
- Please, improve the quality of the figures. Figures and legends must be clearly readable.
We have now improved the quality of the figures to enhance their legibility.
- Authors should revise the citations of the references in the text. In fact, In the text, reference numbers should be placed in square brackets [ ], and placed before the punctuation; for example [1], [1–3] or [1,3].
As suggested, we have now revised the citations of the references.
Reviewer 2 Report
I am grateful to the magazine for the opportunity to review the article entitled “Advertising Innovative Sustainable Fashion: Informational, Transformational, or Sustainability Appeal?”
I find the topic original and topical, but I believe that the authors should make each of the following major changes necessary to increase the quality of the research work.
There is a great need to revise the way of citing throughout the text, as well as the spelling and spelling throughout the paper. In particular in section 2.1
In the Introduction section: More detailed information should be provided on the importance of the paper. For this I recommend the use of the following current references:
- The role of perceived usefulness and annoyance on programmatic advertising: the moderating effect of Internet user privacy and cookies. Corporate Communications: An International Journal, (2022).
- Digitalization in B2B marketing: omnichannel management from a PLS-SEM approach. Journal of Business & Industrial Marketing, (2022).
The literature review section could be updated to more current ones. Even, the justification of the hypotheses aren´t developed and they should be considerably expanded with current references as the following.
- The role of consumer happiness in brand loyalty: a model of the satisfaction and brand image in fashion. Corporate Governance: The International Journal of Business in Society, (2021).
- Promoting social media engagement via branded content communication: A fashion brands study on Instagram. Media and Communication, (2022).
- Influence and relationship between branded content and the social media consumer interactions of the luxury fashion brand Manolo Blahnik. Publications, 9(1), 10. (2021).
The methodology and the sample size seem to me to be correct.
In the conclusion section, the theoretical implications should be expanded upon. I recommend linking the new results to the proposed current benchmarks outlined in the comments above.
Author Response
Dear Reviewer,
We thank you for revising our paper. In the new version of the manuscript, we have addressed all the concerns that you provided for our manuscript. In addition, we have extensively revised the English language and style.
Please, find below a detailed response to your points, with the original comments in italics.
Thank you for your revision.
Best regards,
The Authors
***
I am grateful to the magazine for the opportunity to review the article entitled “Advertising Innovative Sustainable Fashion: Informational, Transformational, or Sustainability Appeal?”
I find the topic original and topical, but I believe that the authors should make each of the following major changes necessary to increase the quality of the research work.
There is a great need to revise the way of citing throughout the text, as well as the spelling and spelling throughout the paper.
As suggested, we have now revised all typos in spelling and in-text citations.
In particular in section 2.1
In the Introduction section: More detailed information should be provided on the importance of the paper. For this I recommend the use of the following current references:
- The role of perceived usefulness and annoyance on programmatic advertising: the moderating effect of Internet user privacy and cookies. Corporate Communications: An International Journal, (2022).
- Digitalization in B2B marketing: omnichannel management from a PLS-SEM approach. Journal of Business & Industrial Marketing, (2022).
The literature review section could be updated to more current ones. Even, the justification of the hypotheses aren´t developed and they should be considerably expanded with current references as the following.
- The role of consumer happiness in brand loyalty: a model of the satisfaction and brand image in fashion. Corporate Governance: The International Journal of Business in Society, (2021).
- Promoting social media engagement via branded content communication: A fashion brands study on Instagram. Media and Communication, (2022).
- Influence and relationship between branded content and the social media consumer interactions of the luxury fashion brand Manolo Blahnik. Publications, 9(1), 10. (2021).
Thanks for your suggestions. We have now updated the literature including all the above references, and revised the text accordingly (lines 57,64; 72-78; 631-648).
The methodology and the sample size seem to me to be correct.
Thank you for your positive feedback on the methods section.
In the conclusion section, the theoretical implications should be expanded upon. I recommend linking the new results to the proposed current benchmarks outlined in the comments above.
As recommended, we have now expanded the implications of the studies in the Discussion section (lines 631-648).
Reviewer 3 Report
Very strange paper - very technical without interpretation and showing why such calculations are needed and what they show, confirm or support. Example - what does mean the confidence interval (-2.43, 5.27), etc as well as so big dispersion especially for "willingness to pay" where arithmetic mean is 44.66 and standard deviation is 51.26. It looks that it is a business project with so big boasting level of themselves. Conclusions are so weak - the weakest I ever saw in scientific paper - there are just some information - report for themselves. In reference list there are indicated only the first letters of the authors - no family name is shown.
Author Response
Dear Reviewer,
We thank you for revising our paper. In the new version of the manuscript, we have addressed all the concerns that you provided for our manuscript. We have also extensively revised the English language and style.
Please, find below a detailed response to your points, with the original comments in italics.
Thank you for your revision.
Best regards,
The Authors
- Very strange paper - very technical without interpretation and showing why such calculations are needed and what they show, confirm or support. Example - what does mean the confidence interval (-2.43, 5.27), etc as well as so big dispersion especially for "willingness to pay" where arithmetic mean is 44.66 and standard deviation is 51.26. It looks that it is a business project with so big boasting level of themselves.
- To reduce the technicality of the paper, we have now better explained the meaning of the analyses conducted (e.g., lines 393-396; 399-400; 415-416;426-428). We have also clarified what the confidence interval means (lines 359-362). In the case of the high standard deviation of the “willingness to pay” variable, it depends on the fact that, when we asked participants to indicate how much they would pay for a bag similar to the one they saw in the post, they were free to indicate any price. The other variables had a more limited dispersion as they were measured on a Likert scale. This information is now better clarified in the Measures section of the paper (lines 342-344).
- Conclusions are so weak - the weakest I ever saw in scientific paper - there are just some information - report for themselves.
- We agree that the conclusions were brief. However, we extensively discussed the results of our paper in the Discussion section, interpreting them with reference to both their theoretical and practical implications (lines 630-648). For this reason, we decided to highlight only the central aspects of what emerged in the Conclusion section. As indicated by the journal guidelines, this section is not mandatory but can be added to the manuscript if the discussion is unusually long or complex. Anyhow, we have now extended this section of the paper (lines 657-663).
- In reference list there are indicated only the first letters of the authors - no family name is shown.
- We have now completely revised the reference list of the paper.
Reviewer 4 Report
Dear authors
The paper looks fine and discusses the importance of sustainable fashion. However, there are some typo errors and it should be addressed.
For instance, in the abstract, you mentioned PVD, what is it?
The citing reference style is not followed what sustainability journal ((Eder-Hansen et al., 2017;)
Conclusion part should be improved
Author Response
Dear Reviewer,
We thank you for agreeing to review our paper. In the revised version of the manuscript, we have addressed all the concerns that you provided for our manuscript. We have also extensively revised the English language and style.
Please, find below a detailed response to your points, with the original comments in italics.
Thank you for your revision.
Best regards,
The Authors
- Dear authors
The paper looks fine and discusses the importance of sustainable fashion. However, there are some typo errors and it should be addressed.
For instance, in the abstract, you mentioned PVD, what is it?
Thank you for your positive feedback. We have now checked all the typos and avoided the PVD acronym in the abstract.
- The citing reference style is not followed what sustainability journal ((Eder-Hansen et al., 2017;)
We have now carefully revised the citing reference style.
3.Conclusion part should be improved
We agree that the conclusions were brief. However, we extensively discussed the results of our paper in the Discussion section, interpreting them with reference to both their theoretical and practical implications (lines 630-648). For this reason, we decided to highlight only the central aspects of what emerged in the Conclusion section. As indicated by the journal guidelines, this section is not mandatory but can be added to the manuscript if the discussion is unusually long or complex. Anyhow, we have now extended this section of the paper (lines 657-663).
Round 2
Reviewer 2 Report
The co-authors have improved the paper according to the comments made in the last revision.
Author Response
Thank you very much for your positive feedback.
Best regards,
The authors
Reviewer 3 Report
The paper is improved, but still there are some suggestions:
*) more explanations for several very wide confidence intervals;
*) conclusions are to short and also as report on done - they do not reflect main findings and conclusions - there are much more findings which are not reflected in the conclusions;
*) use the same reference preparation style for all sources in reference list, for example, very different are number 4; 15; 24; 25; 46
Author Response
Dear Reviewer,
Thank you again for your availability in revising our manuscript.
As recommended, we have now commented on the wide confidence intervals in the limitation section of the manuscript (lines 583-592). We have also better detailed the main findings in the conclusions (lines 638-665). Finally, we have revised the references’ style.
Best regards,
The Authors